# Antileukemia Activity and Mechanism of Platinum(II)-Based Metal Complexes

**DOI:** 10.3390/molecules27249000

**Published:** 2022-12-17

**Authors:** Maria Letizia Di Pietro, Claudio Stagno, Thomas Efferth, Ejlal A. Omer, Valeria D’Angelo, Maria Paola Germanò, Anna Cacciola, Federica De Gaetano, Nunzio Iraci, Nicola Micale

**Affiliations:** 1Department of Chemical, Biological, Pharmaceutical and Environmental Sciences, University of Messina, 98166 Messina, Italy; 2Department of Pharmaceutical Biology, Institute of Pharmaceutical and Biomedical Sciences, Johannes Gutenberg University, 55128 Mainz, Germany; 3“Antonio Imbesi Foundation”, Piazza Pugliatti 1, 98122 Messina, Italy

**Keywords:** platinum(II)-based complexes, anticancer agents, leukemia cell lines, CAM assay, molecular modeling studies

## Abstract

Transition metal complexes have continued to constitute an appealing class of medicinal compounds since the exceptional discovery of cisplatin in the late 1960s. Pt(II)-based complexes are endowed with a broad range of biological properties, which are mainly exerted by targeting DNA. In this study, we report a significant biological investigation into and computation analyses of four Pt(II)-complexes, namely, **LDP-1–4**, synthesized and characterized according to previously reported procedures. Molecular-modelling studies highlighted that the top two **LDP** compounds (i.e., **LDP-1** and **LDP-4**) might bind to both matched and mismatched base pair sites of the oligonucleotide 5′-(dCGGAAATTACCG)_2_-3′, supporting their anticancer potential. These two complexes displayed noteworthy cytotoxicity in vitro (sub-micromolar–micromolar range) against two leukaemia cell lines, i.e., CCRF-CEM and its multi-drug-resistant counterpart CEM/ADR5000, and remarkable anti-angiogenic properties (in the sub-micromolar range) evaluated in an in vivo model, i.e., a chick embryo chorioallantoic membrane (CAM) assay.

## 1. Introduction

Since the serendipitous discovery of cisplatin cytotoxicity by Rosenberg, which was first reported in 1969 [1], many platinum(II)-based metal complexes with two inert ligands and two labile ones in the *cis* position have been synthesized and tested as potential antitumor drugs. To achieve its anticancer activity, once cisplatin enters the tumour cell, it must first undergo the replacement of a chloride by a water molecule, followed by the coordination of the platinum to a purine base. Afterwards, through the replacement of the second chloride with a water molecule and its binding to another purine base, a bifunctional intrastrand or interstrand adduct is formed, which causes the DNA helix to distort toward the major groove [2]. The subsequent failure of the DNA repair systems is important for the apoptosis of the tumour cell [3].

However, despite the thousands of compounds tested, very few Pt(II) complexes have been accepted as therapeutic drugs. Therefore, the research in the field of metal complexes has also been extended to: (i) platinum(II)-based complexes with different structures (therefore with a possible different mode of action) [4], such as phenanthriplatin, which covalently binds to DNA forming a monofunctional adduct and that seems to be more active than cisplatin and to show a spectrum of activity significantly different from that of the classical anticancer platinum-based agents [5,6]; (ii) compounds of transition metals other than platinum [7]; and (iii) the development of drug delivery systems with which to transport the platinum-based chemotherapeutic compounds and release them only in proximity to cancer cells, so as to reduce the serious side effects associated with the administration of classical platinum-based drugs [8,9,10].

Even the non-covalent interaction of a species with DNA can be exploited in the chemotherapeutic field [11,12,13,14,15,16]. Metal complexes with extended aromatic ligands have been widely used throughout the years due to their capability to interact non-covalently with nucleic acids. Their aromatic moiety, in fact, can intercalate between base-pairs, being stabilized by aromatic π–π stacking and eventually proceeding to interfere with the opening of the helix, which is the basis of DNA duplication and transcription processes [17,18].

In this paper, we report the results of a cytotoxicity study carried out on two leukaemia cell lines of four Pt(II) metal complexes (encoded as “**LDP-1-4**” and shown in Figure 1) capable to interact with DNA both non-covalently—by intercalating their aromatic coordination ligand to the metal centre—and/or by coordination with purine nucleobases. Earlier studies have shown that the intercalation of the extended aromatic ligand of these complexes occurs into the DNA double helix when it is coordinated to the central Pt(II) ion in cationic complexes in which the remaining ligands were pyridines instead of chlorides [19,20], as well as the possibility of using Pt(II) complexes with 1,10-phenanthroline (phen) and phen-derivatives as antitumor and antimicrobial agents [21,22].

Therefore, this dual mode of interaction could allow such compounds to initially interact with DNA by intercalating the aromatic moiety between adjacent nucleobases and then, once the metal centre is close to the biopolymer, promote the coordination of platinum to DNA bases. As shown in Figure 1, while the complex with the phen ligand (**LDP-1**) and the one with its derivative dipyrido [3,2-a:2′,3′-c]phenazine (dppz) (**LDP-2**) show the usual Pt(II) square-planar geometry, the other two complexes possess an unusual L-shaped geometry, which is due to the coordination of the 6,7-dimethyl-2,3-bis(2-pyridyl)quinoxaline (DMeDPQ) (**LDP-3**) and 2,3-bis(2-pyridyl)benzo[*g*]quinoxaline (BDPQ) (**LDP-4**) ligands. Such ligands, in fact, are known to bind Pd(II) and Pt(II) ions, forming a seven-membered ring through both pyridine N atoms in a *cis* conformation, rather than a five-membered ring using one pyridine N and one pyrazine N atom [20,23,24]. The anticancer properties of the two square-planar-type Pt(II) complexes have been previously assessed against a panel of solid tumour cell lines. Specifically: A549 (for **LDP-1**) [21], A498, EVSA-T, H226, IGROV-1, M19-MEL, MCF-7, WIDR, A2780, and A2780R (for **LDP-2**) [22].

In this work, all the compounds have been synthesised and characterised as previously reported [20,24,25]. Then, they underwent substantial biological assessments to further demonstrate their potential usefulness as anticancer agents. These assessments include an in vitro cytotoxicity evaluation of the Pt(II) complexes against two leukaemia cell lines and in vivo angiogenesis studies via a chick embryo chorioallantoic membrane (CAM) model. Finally, molecular modelling studies were carried out in order to shed light on the molecular recognition differences between the square-planar geometry of **LDP-1** and the L-shaped geometry of **LDP-4** with the expected biological target, i.e., DNA.

## 2. Results and Discussion

### 2.1. Biological Assessments

In line with our recent research activity dealing with the development of transition metal-based complexes as potential anticancer agents [26,27,28], we selected two leukaemia cell lines, namely, the drug-sensitive CCRF-CEM cell line and its multidrug-resistant sub-cell line CEM/ADR5000, for the assessment of the cytotoxicity of our set of Pt(II) compounds. The resazurin reduction assay was used to pre-test the four Pt(II) compounds at a fixed concentration of 10 µM against drug-sensitive CCRF-CEM cells. Figure 2 shows the results ordered according to their decreasing cell viability (in a waterfall plot). As can be seen, all the compounds showed inhibitory activity below the cut-off point of 30% cell viability. Detailed results regarding the percentage of residual cell viability of the CCRF-CEM cells are reported in Table 1.

The top two compounds, one for each structural motif, i.e., the L-shaped **LDP-4** and the square-planar **LDP-1**, were selected for further assessments, including IC_50_ determination, resistance ratio calculations, and selectivity index (SI), for which the latter was evaluated using human peripheral blood mononuclear cells (PBMCs). The dose response curves of the two selected compounds plus the control drug cisplatin are shown in Figure 3.

Based on these dose–response curves, 50% inhibition concentrations (IC_50_) were calculated (Table 2). **LDP-1** displayed antiproliferative activity in the low-micromolar range against both leukaemia cell lines, whereas **LDP-4** showed activity in the sub-micromolar range against CCRF-CEM cells and micromolar values against CEM/ADR5000 cells. Noteworthily, both complexes were revealed to be more active than the reference drug cisplatin against the drug-sensitive leukemic cells (i.e., cisplatin→IC_50_ = 5.82 μM vs. IC_50_ = 1.71 μM and IC_50_ = 0.82 μM for **LDP-1** and **LDP-4**, respectively). Then, the IC_50_ values were used to calculate the degrees of resistance. The degrees of resistance were 2.32- and 21.29-fold higher for **LDP-1** and **LDP-4**, respectively, compared to cisplatin, which showed a value of 0.56. In regard to the SI towards healthy cells, **LDP-1** was revealed to be even more cytotoxic to normal PBMCs (i.e., IC_50_ = 0.82 μM for PBMCs vs. IC_50_ = 1.71 μM and IC_50_ = 3.97 μM for CCRF-CEM and CEM/ADR 5000, respectively), whereas **LDP-4** presented an IC_50_ value towards PBMCs that was between those of the two leukemic cells (2.03 μM for PBMCs vs. 0.82 μM and 17.46 μM for CCRF-CEM and CEM/ADR 5000, respectively). Therefore, only the L-shaped derivative **LDP-4** shows a moderate degree of tumour specific inhibition (SI = 2.5 calculated towards CCRF-CEM cells).

As leukaemia is associated with a high rate of metastasis (which primarily affects the hematopoietic organs) [29], and since vascularity and the levels of angiogenic factors are directly related to the leukemogenic process [30,31,32], we were also determined to exploit the chick embryo chorioallantoic membrane (CAM) assay as an in vivo model to further investigate the therapeutic potential of this type of compound. The CAM assay is an efficient and cost-effective model that is generally employed to study the vascular effects of new therapeutic agents [33]. However, it has also been successfully used as a predictive model in acute toxicological studies [34], as a patient-derived xenograft platform for preclinical oncologic research [35], for studying human blood cell malignancies [36], and for invasion/metastasis studies [37,38].

The anti-angiogenic activity of **LDP-1** and **LDP-4** in the CAM model is reported in Figure 4. The effects were evaluated using a dose–response curve and expressed in terms of IC_50_ values. The results showed that both compounds **LDP-1** and **LDP-4** exhibit a better anti-angiogenic response (IC_50_ = 0.87 μM ± 0.04 and 0.77 μM ± 0.03, respectively) compared to retinoic acid (IC_50_ = 2.6 μM ± 0.04), which was used as a reference compound.

Selected photomicrographs of the CAMs highlighting the anti-angiogenic properties of both Pt(II) complexes are reported in Figure 5. The control group has a rich vascular network. Conversely, it was clear that **LPD-1** and **LPD-4** drastically reduced the microvasculature of the CAMs, inducing a thinning of the vessels in the area of application as compared to the positive control.

As it has been recognised that increased angiogenesis and angiogenic factors play a significant role in the course and disease process of leukaemia, the anti-angiogenic effects of **LDP-1** and **LDP-4** in the in vivo model seem to generate new fundamental insights required for the design of metal-based complexes for improved anti-leukaemia therapy.

### 2.2. Molecular-Modelling Studies

Two DNA regions were considered for molecular docking studies, which were performed to ascertain the potential binding poses of **LDP-1** and **LDP-4**, using the crystallographic model of the DNA duplex (PDB ID: 4E1U) in complex with the **[Ru(bpy)_2_dppz]^+2^** metal complex as a docking target [39]. We considered two binding regions: the first one formed by the well-matched DNA base pairs A6-T19 and T7-A18 (BP1—binding pocket 1) and the second one lined by the well-matched DNA base pairs G3-C22 and A5-T20 and by the mismatched A4-A21 base pairs (BP2—binding pocket 2).

The docking studies suggest that **LDP-1** and **LDP-4** might interact with DNA bases from both BP1 and BP2, overlapping with the experimental bound conformation of the dipyrido[3,2-a: 2′,3′-c]phenazine moiety of **[Ru(bpy)_2_dppz]^+2^** (Figure 6A–D). It is worth noting that the square-planar complex **LDP-1** shows a slightly different binding mode compared to the L-shaped complex **LDP-4**. Indeed, the molecular plane of **LDP-1** is rotated by about 180° along its orthogonal axis, placing the metal toward the centres of BP1 and BP2 (Figure 6A,C). On the other hand, **LDP-4** intercalates its aromatic portion into the base pairs of both BP1 and BP2, with its metal facing the outside of the minor groove (Figure 6B,D). This different behaviour is likely due to two main reasons: (i) L-shaped complexes, because of their geometry, cannot centre the metal atom into BP1 and BP2 due to steric hindrance, and (ii) because of the smaller size of **LDP-1** compared to **LDP-4**, with the latter one establishing improved π–π stacking interactions with the nucleobases by its differently shaped aromatic surface.

The docking-predicted binding poses of **LDP-1** and **LDP-4** were then challenged by molecular dynamics (MD) simulations. Even though **LDP-1** intercalates the BP1 bases A6, T7, A18, and T19, its RMSD is fairly unstable during the MD simulation (Figure 7A) and it was found to be fairly free with respect to rearranging its pose. Anyway, it is worth nothing that, for the majority of the MD simulation time, **LDP-1** remained in the major groove, similar to cisplatin [40], with its metal atom close to A6, T7, A18, and T19. In contrast, the binding of **LDP-1** to BP2 was stable (Figure 7C). It docks deeper into BP2, stacking with G3, A5, T20, and C22, with its metal atom placed at the minor groove, far from the nucleobases.

As shown in Figure 7B,D, the small RMSD variations recorded during the MD simulations highlight the stability of **LDP-4** binding to both BP1 and BP2. **LDP-4** binds to BP1, mainly interacting by π–π stacking with A6, T7, A18, and T19. The binding of **LDP-4** to BP1 looks significantly more stable than for **LDP-1** because of its different geometry (L-shaped) and aromatic surface shape, which is in sharp contrast to the square-planar **LDP-1** complex.

The binding of **LDP-4** to BP2 causes a slight rearrangement of the binding site; indeed, the complex loses contact with C22 and shifts toward G3 and A5, followed by the movement of T20. After this rearrangement, **LDP-4** stacks tightly with G3, A4, A5, and T20, with its metal atom lying next to A21.

## 3. Materials and Methods

### 3.1. Cell Lines and Compounds

Human CEM-CCRF and CEM/ADR5000 leukaemia cell lines were cultured in RPMI-1640 medium supplemented with 10 % foetal bovine serum and 1% penicillin/streptomycin (Invitrogen, Darmstadt, Germany). The cells were incubated in a humidified atmosphere of 5% CO_2_ in air at 37 °C. The four Pt(II)-based compounds were prepared as 20 mM stock solutions in DMSO and then diluted 200-fold with a medium. The stock solutions were stored at −20 °C.

### 3.2. Cell Proliferation Inhibition Assay

The proliferation inhibition activity of the four compounds was detected using the resazurin assay. CEM-CCRF were first exposed to the test compounds at a fixed concentration of 10 μM. For IC_50_ determination of the selected compounds, 10 concentrations were prepared for each of the compounds in a range of 0.3–100 μM. Both CEM-CCRF and CEM/ADR5000 suspension cells were treated immediately after seeding. After 72 h incubation, 20 μL of 0.01 % resazurin (Promega, Mannheim, Germany) was added to each well. Resazurin fluorescence was measured after 4 h incubation using an Infinite M2000 Pro plate reader (Tecan, Crailsheim, Germany) at Ex/Em = 550 nm/590 nm wavelength [41,42]. Cell viability was calculated in comparison to DMSO control. The final concentration of DMSO was 0.5 %. Cisplatin was used as positive control. This experiment was repeated three times with one of the six wells for each concentration. Cell viability was calculated in comparison to DMSO control.

### 3.3. Toxicity in Normal Cells

Peripheral blood (PB) was collected from a healthy donor in a plastic Monovette EDTA tube. The isolation of human peripheral mononuclear cells (PBMCs) was executed using Histopaque^®^ (Sigma-Aldrich, St. Louis, MO, USA), as previously described [43]. Consequently, 3 mL of blood was carefully layered over 3 mL Histopaque^®^ and centrifuged at 400× *g* for 30 min at RT. The layer containing PBMCs at the interface between blood serum and Histopaque^®^ was transferred into a new tube and washed with PBS three times. The isolated cells were suspended in Panserin 413 medium (PAN-Biotech, Aidenbach, Germany) supplemented with 2.5% phytohemagglutinin M (PHA-M, Life Technologies, Darmstadt, Germany). Finally, the cell viability was measured using resazurin assay as described above.

### 3.4. CAM Assay

The chick embryo CAM assay was performed following the modified method of Bader et al. [44]. Fertilised chicken eggs purchased from a local supplier were maintained in a humidified incubator at 37 °C. The eggs were placed in a horizontal position and rotated after 24 h. After 4 days of incubation, a window (1–2 cm large) was created on the egg to assess the state of the embryonic blood vessels. The embryos’ development was checked by visual inspection. Eggs with dead embryos or minor vascularization were excluded. **LPD-1** and **LPD-4** (0.1–1.0 µM) were dissolved in DMSO (0.2% *v*/*v*) with Tris buffer (pH 7.4). Retinoic acid (1–3 µM/egg) was used as a positive control. Six eggs were used for each group. At the end of the incubation period (48 h), each egg was observed under a stereomicroscope (SMZ-171 Series, Motic, Hong Kong, China) to visualise the CAM’s microvasculature. The images were acquired using a digital camera (Moticam^®^ 5 plus). The anti-angiogenic effects were evaluated by counting the total number of junctions in a standardised area of each CAM using artistic software (Paint.ink). The percentage of inhibition was calculated using the following equation:Inhibition (%) = (A − B/A) 
where: A = total number of junctions in control CAM;

B = total number of junctions in treated CAM.

Finally, we calculated the IC_50_ values from the dose–response curves of compounds and retinoic acid.

### 3.5. Molecular-Modelling Studies

The crystal structure of DNA duplex 5′-(dCGGAAATTACCG)2-3′ crystallised with inhibitor **[Ru(bpy)_2_dppz]^+2^** (PDB ID: 4E1U) [39] was downloaded from the protein data bank [45] and prepared by means of AutoDockTools 1.5.6 [45] and used as DNA duplex target. Two docking grids were generated through AutoGrid 4.2.6 [46] centred on the experimental BP1- and BP2-bound conformations of **[Ru(bpy)_2_dppz]^+2^**. Grids’ dimensions were set to 60 points on each axis, with a grid spacing of 0.375 Å. Metal complexes were sketched using Maestro GUI [47] and optimised through 100 steps of B3LYP DFT calculations, using LACVP* basis set for platinum and 6-31G* for all the other atoms.

Docking simulations were performed using AutoDock 4.2.6 [46] and parameters for platinum were added to the default parameters (atom_par Pt 2.75 0.080 12.000 -0.00110 0.0 0.0 0 -1 -1 4 # Non H-bonding). For each ligand, 100 genetic algorithm runs were executed. Population size was set to 150 individuals, the maximum number of energy evaluations was set to 2,500,000, and mutation and crossover rates were set to 0.02 and 0.8, respectively. Clustering of docking poses was performed using an RMSD cut-off of 2.0 Å, and the clusters were finally ranked by the lowest binding energy for each cluster. The best-scoring docking poses for DNA-BP1/**LDP-1**, DNA-BP1/**LDP-4**, DNA-BP2/**LDP-1,** and DNA-BP2/**LDP-4** were then submitted for MD simulations, which were set up and run using Desmond [48]. **LDP-1** and **LDP-4** partial charges were retrieved from the DFT calculations, solvation was treated implicitly using the TIP3P water model [49], and 22 Na^+^ ions were added to neutralize the system, for which OPLS_2005 was used as force-field [50]. Prior to the production stage, the four systems were relaxed as previously reported [51]. After systems’ relaxation, 480 ns long simulations were run at a temperature of 300 K in the NPT ensemble using a Nose–Hoover chain thermostat and Martyna–Tobias–Klein barostat (1.01325 bar). Time steps were set to 2 fs, 2 fs, and 6 fs for bonded, near, and far interactions, respectively. Recording intervals for trajectories were set to 480 ps. With the exception of residues A6, T7, T18, and T19 for BP1-bound complexes, and G3, A4, A5, T20, A21, and C22 for BP2-bound systems, non-H atoms were constrained by 1 kcal/mol. **LDP-1** and **LDP-4** RMSD calculations over the MD trajectories were performed using the DNA structure for superimposition. Open-source PyMOL v. 1.8.4.0 was used for visual inspection and to create molecular representations.

## 4. Conclusions

In summary, four Pt(II) complexes (**LDP-1-4**), synthesised and characterised according to previous methods, were taken into account for an exhaustive investigation of their role as potential antileukemia agents. After a preliminary in vitro screening at 10 μM on two leukaemia cell lines, the top two compounds, i.e., **LDP-1** and **LDP-4**, were selected for detailed studies, which encompassed further biological assessments and computational analyses. From the cell-based assay, it emerged that both compounds are highly cytotoxic towards the two selected leukaemia cells (drug-sensitive CCRF-CEM and multi-drug-resistant CEM/AD5000 cells) with IC_50_ values in the sub-micromolar–micromolar range. Moreover, they displayed superior cytotoxicity towards the drug-sensitive leukemic cells as compared to the reference drug cisplatin. **LDP-4** also showed a certain degree of tumour specificity (SI = 2.5 on PBMC cells), whereas **LDP-1** displayed a better cross-resistance profile (DS = 2.32). In addition, both compounds showed notable anti-angiogenic properties (IC_50_ 0.77–0.87 μM) assessed in vivo by CAM assays. The molecular-modelling studies highlighted the potential capabilities of both square-planar and L-shaped complexes to bind to both matched and mismatched base pair sites of the oligonucleotide 5′-(dCGGAAATTACCG)_2_-3′, although with substantial differences. Indeed, the L-shaped complexes seem to be unable to steer their metal atom toward the centre of the two sites because of steric hindrance. Moreover, **LDP-4**, which is endowed with a more extended aromatic structure, interacts with the nucleobases via π–π stacking to a greater extent than **LDP-1**.

## Figures and Tables

**Figure 1 molecules-27-09000-f001:**
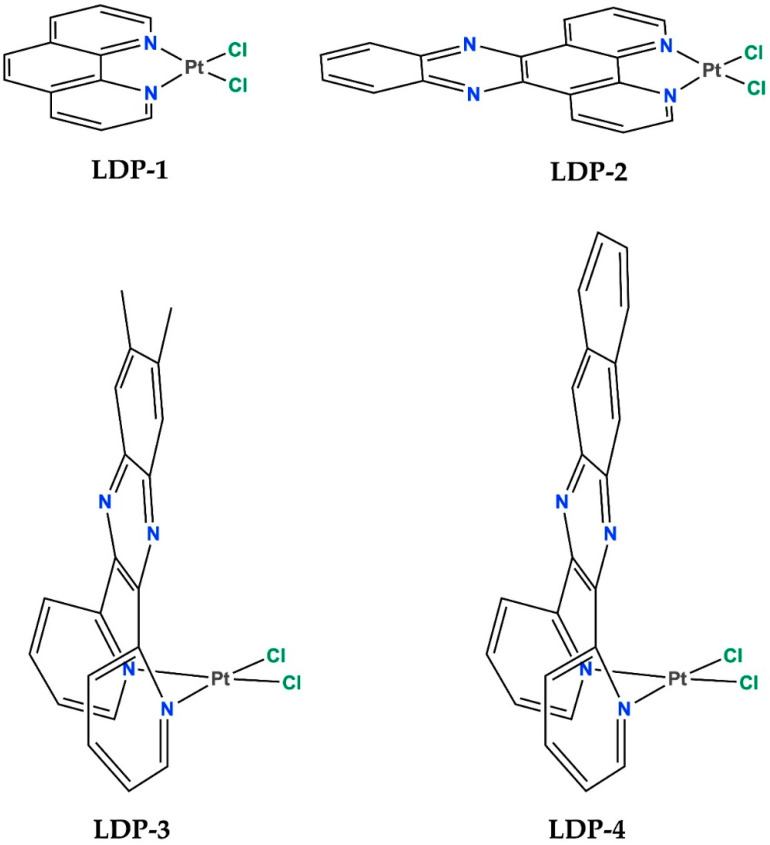
Perspective scheme of the Pt(II) complexes LDP-1–4.

**Figure 2 molecules-27-09000-f002:**
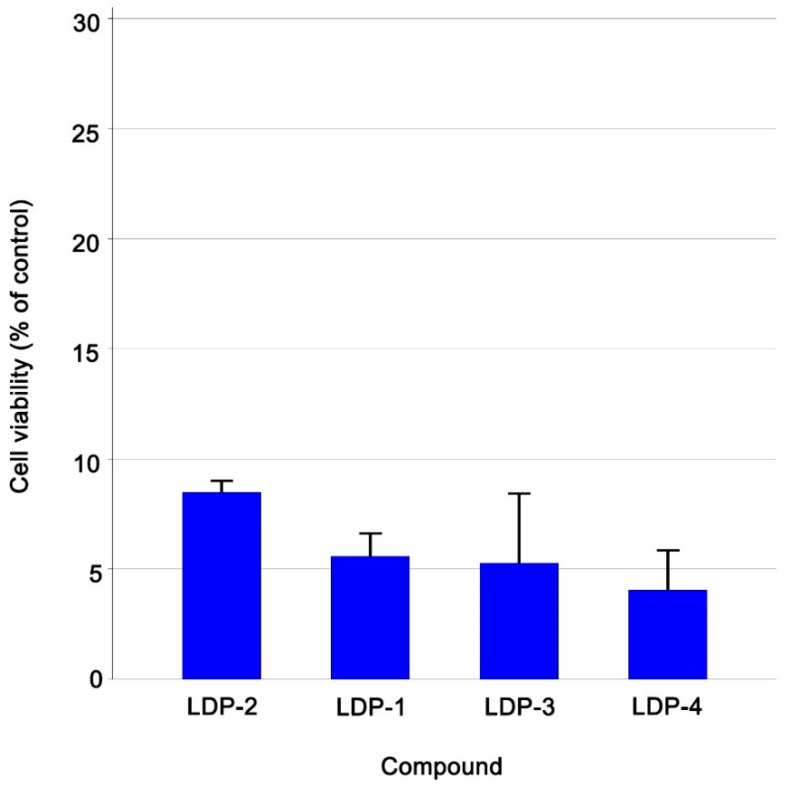
Cytotoxicity of the four Pt(II) compounds towards CCRF-CEM leukaemia cells at a fixed concentration of 10 µM as measured by the resazurin reduction assay. All data are presented as mean ± SE of three independent experiments.

**Figure 3 molecules-27-09000-f003:**
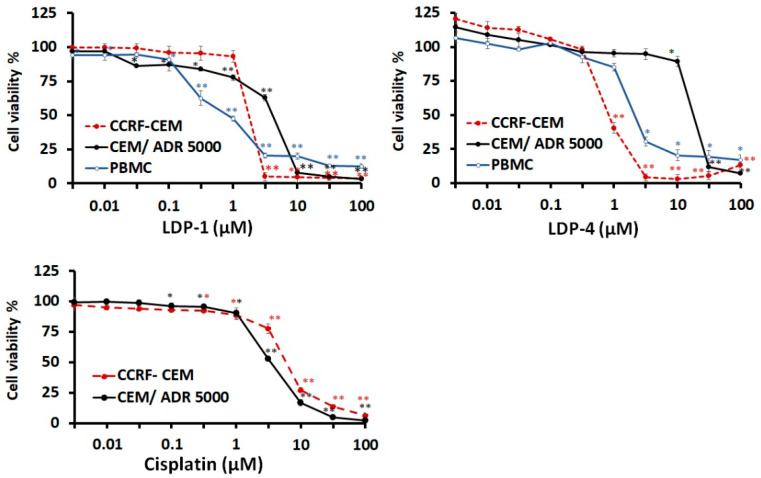
Cytotoxicity of the top two selected compounds towards drug-sensitive parental CCRF-CEM tumour cells and their P-glycoprotein (MDR1/ABCB1)-expressing, multidrug-resistant subline, CEM/ADR5000, as determined by resazurin assays. In addition, human peripheral mononuclear cells (PBMCs) were investigated as normal counterparts to the leukaemia cell lines. Cisplatin was used as positive control drug to verify the multidrug resistance phenotype of the CEM/ADR5000 cells. All data are presented as mean ± SE of three independent experiments. One asterisk (*) indicates *p*-value less than 0.05 (*p* < 0.05); two asterisks (**) indicate *p*-value less than 0.01 (*p* < 0.01).

**Figure 4 molecules-27-09000-f004:**
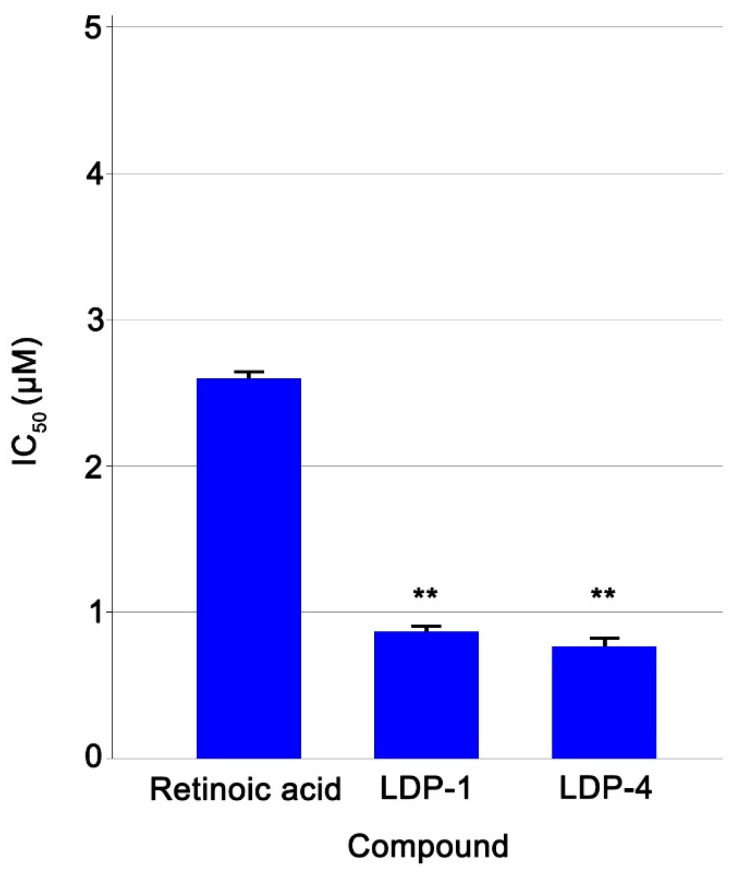
The anti-angiogenic activity of compounds and retinoic acid (positive control) was calculated as inhibition percentage versus negative control (100% of angiogenic activity) in a set of experiments (n = 6). IC_50_ values were then calculated, representing the concentration that caused 50% of the angiogenic activity. The statistical significance was evaluated by one-way analysis of variance (ANOVA) followed by Student’s test. *p* < 0.01 was considered statistically significant. ** *p* < 0.01 vs. retinoic acid.

**Figure 5 molecules-27-09000-f005:**
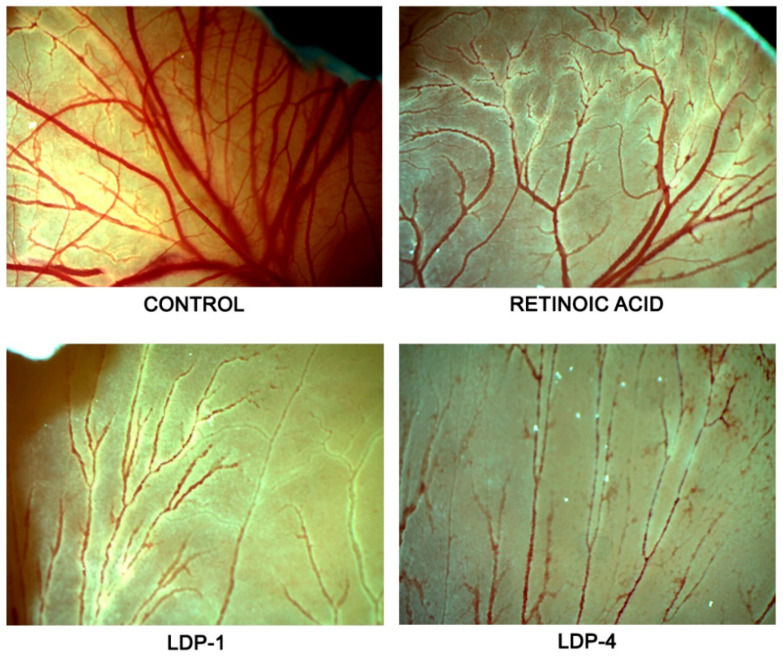
Representative photomicrographs of the chick embryo chorioallantoic membranes (CAMs) treated with **LPD-1** and **LPD-4** (0.5 μM). Retinoic acid was used as positive control (3 μM). The images of the CAMs were captured using a stereomicroscope (SMZ-171 Series, Motic, San Antonio, TX, USA) equipped with a digital camera (Moticam^®^ 5 plus, Motic, San Antonio, TX, USA).

**Figure 6 molecules-27-09000-f006:**
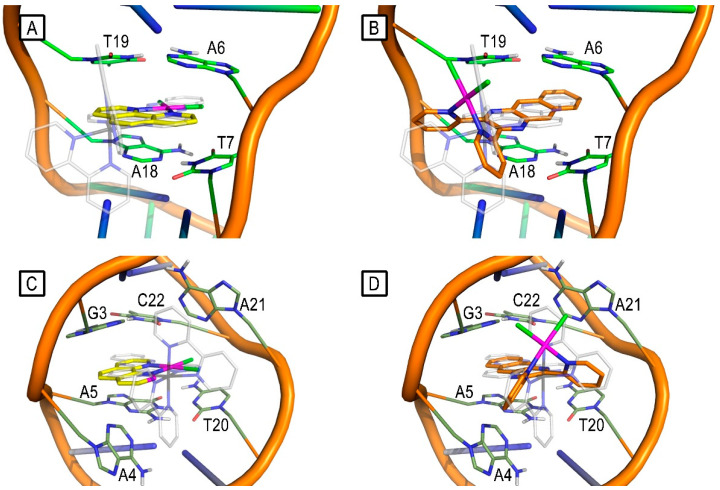
(**A**) Docking pose of **LDP-1** (yellow sticks) bound to BP1 (green sticks). (**B**) Docking pose of **LDP-4** (orange sticks) bound to BP1 (green sticks). (**C**) Docking pose of **LDP-1** (yellow sticks) bound to BP2 (olive sticks). (**D**) Docking pose of **LDP-4** (orange sticks) bound to BP2 (olive sticks). DNA phosphodiester backbone is represented by orange cartoons. **[Ru(bpy)_2_dppz]^+2^** experimental position is depicted for reference, in every panel, as white transparent sticks.

**Figure 7 molecules-27-09000-f007:**
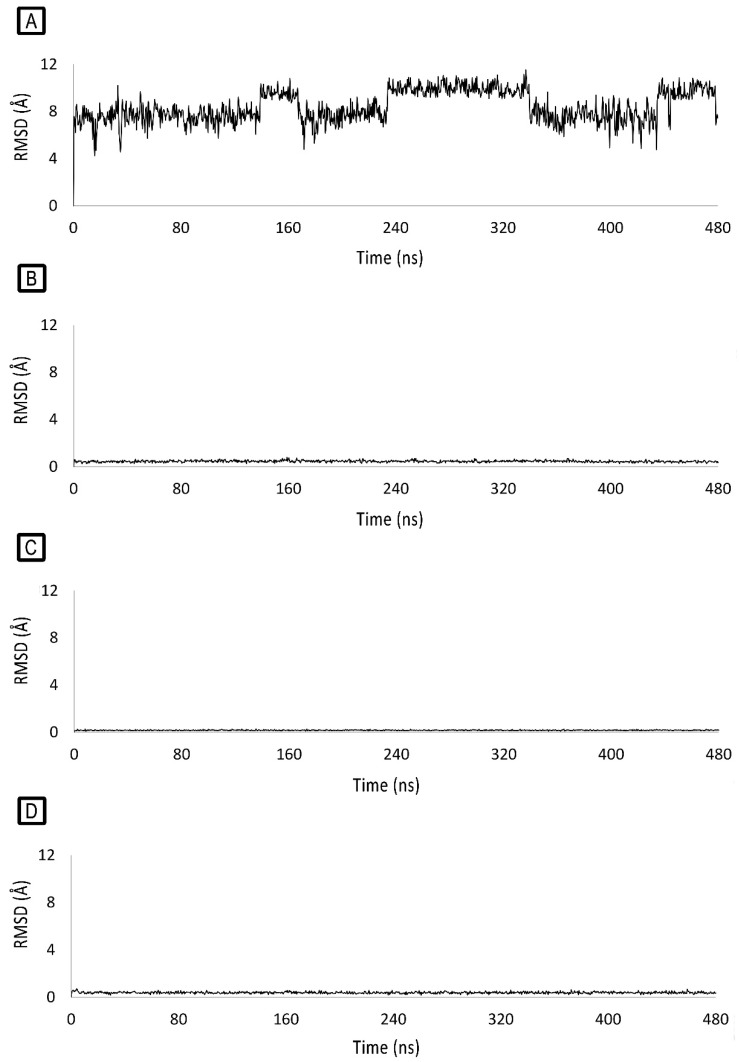
RMSD variations over 480 ns-long MD simulations of (**A**) BP1-bound **LDP-1**; (**B**) BP1-bound **LDP-4**; (**C**) BP2-bound **LDP-1,** and (**D**) BP2-bound **LDP-4**.

**Table 1 molecules-27-09000-t001:** Data regarding the screening at 10 μM against CCRF-CEM leukaemia cells reported as residual cell viability %.

Compound Name	Cell Viability % (±Sd) Ccrf-Cem Cells
**LDP-1**	5.57 ± 1.08
**LDP-2**	8.49 ± 0.59
**LDP-3**	5.28 ± 3.18
**LDP-4**	4.05 ± 1.84

**Table 2 molecules-27-09000-t002:** Cytotoxicity of **LDP-1** and **LDP-4** towards drug-sensitive CCRF-CEM, multidrug-resistant CEM/ADR5000, and healthy PBMC cells using the resazurin reduction assay. All values are shown as mean ± standard deviation (SD) of three independent experiments. The degree of resistance was calculated by dividing the IC_50_ value of resistance by that of the sensitive cells.

Compound Name	CCRF-CEM		CEM/ADR5000		PBMC		Degree of Resistance
	IC_50_ (µM)	SD	IC_50_ (µM)	SD	IC_50_ (µM)	SD	
**LDP-1**	1.71	0.05	3.97	0.16	0.82	0.1	2.32
**LDP-4**	0.82	0.06	17.46	0.96	2.03	0.06	21.29
**Cisplatin (µM)**	5.82	0.16	3.28	0.41	-	-	0.56

## Data Availability

The data that support the findings of this study are available from the corresponding author upon reasonable request.

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
