# Peer review of "Antileukemia Activity and Mechanism of Platinum(II)-Based Metal Complexes"

_molecules, 2022, doi:10.3390/molecules27249000_

Round 1
Reviewer 1 Report (Previous Reviewer 1)
The paper is clear and well written. I have no comments.
Author Response
Reviewer #1
Comments and Suggestions for Authors.
The paper is clear and well written. I have no comments
- Once again, we thank the Reviewer #1 for the appreciation of our work.
Reviewer 2 Report (Previous Reviewer 3)
Authors have revised the manuscript in better manner, but authors are suggested to improve the quality of figures and graph. After this the manuscript can be considered for publication.
Author Response
Reviewer #2
Comments and Suggestions for Authors
Authors have revised the manuscript in better manner, but authors are suggested to improve the quality of figures and graph. After this the manuscript can be considered for publication.
- We thank the Reviewer #2 and accept his/her suggestions to further improve the quality of the article. All figures and graphs have been revised and/or improved in terms of quality. Specifically:
Figure 1: Revised (quality improvement) and caption modified indicating “perspective scheme” of the Pt(II) complexes as we drew the structures trying to highlight their geometry, i.e. square-planar for LDP-1 and LDP-2, and L-shaped for LDP-3 and LDP-4.
Figure 2: Histogram properly revised and aligned with that of Figure 4 in terms of quality and style.
Figure 3: Resolution improved.
Figure 4: As for Figure 2 (see above)
Figure 5. Resolution improved for all 4 photomicrograps.
Figure 6: Resolution improved
Figure 7: Resolution improved
This manuscript is a resubmission of an earlier submission. The following is a list of the peer review reports and author responses from that submission.
Round 1
Reviewer 1 Report
This paper deals with Pt (II) complexes. The paper is clear and well written.
There are some errors in this work:
1. Lack of references in preliminary [34-40, p.1; 46-50, p. 2]. References to be completed in the publication
doi:10.4061/2010/201367
doi: 10.1016/B978-0-12-409547-2.14251-9
doi: 10.3390/ijms22179264
doi: 10.1016/j.mattod.2015.05.017
2. Please mention the action of phenantriplatin, the structure of which is similar to the compounds presented by you.
3. Please comment in the publication why leukemia was selected for testing platinum compounds.
4. It is a pity that the obtained results were not compared to free cisplatin.
Reviewer 2 Report
The manuscript by Di Pietro et al. deals with anticancer activity of several heteroleptic Pt(II) complexes of general formula LPtCl2. Such complexes are widely used in chemotherapy of cancer so, in general, this entry is worth being published.
However, we do think that this work is more suitable for another, more medicine-oriented journal, such as Pharmaceutics.
Reviewer 3 Report
The current research article has addressed Anticancer Activity and Mechanism of Platinum(II)-based Metal Complexes via in vitro and in silico approaches. The manuscript comprises very limited experimental studies related to antitumor activity and its mechanistic approach. The theme is interesting; however, the manuscript lacks proper study design and has significant faults related to formatting/structure of manuscript that the authors should clarify before a new peer-revision round. Some of my comments are below:
1. Authors have designed the anticancer activity of selected compounds but only two leukemia cell lines were selected. If authors want to demonstrate the anticancer activity then more cancer cell lines should be selected or else title should be specific to leukemia.
2. I have serious concern with the structure of the manuscript, for example authors have performed only two in vitro assays to explore the anticancer potential which are not sufficient similar in case of toxicity studies. These assays revealed very limited information regarding the anticancer and mechanism of platinum-based compounds.
3. Similarly in section 3.4 authors have performed the Cam assay to demonstrate the antiangiogenic potential, however there are specific assay which can confirm the same in better manner. Authors must include specific assays.
4. Authors have also performed the in silico experiments which are not presented in a very well manner.
4. No statistical analysis was described in methodology section. There is no representation of statistical significant (p-value) compared to control and control is missing from graph. This is a serious flaw in my opinion.
5. Very poor presentation of graph and diagram. The quality and resolution of each graph and diagram is not up to the mark.
6. I am unable to connect section or assays used with each other.
Altogether, this research article in context to journal’s criteria lacks quality and title did not match with the content. In this manuscript, main problem is with the selection of bioassays which is not aligned with each other. Therefore, the manuscript is not suitable for publication; it should be carefully revised before the following peer-review process.